# Vacuum-field-induced state mixing

Diego Fernández de la Pradilla⋆, Esteban Moreno and Johannes Feist

Departamento de Física Teórica de la Materia Condensada
and Condensed Matter Physics Center (IFIMAC),
Universidad Autónoma de Madrid, E-28049 Madrid, Spain

⋆ diego.fernandez@uam.es

## Abstract

By engineering the electromagnetic vacuum field, the induced Casimir-Polder shift (also known as Lamb shift) and spontaneous emission rates of individual atomic levels can be controlled. When the strength of these effects becomes comparable to the energy difference between two previously uncoupled atomic states, an environment-induced interaction between these states appears after tracing over the environment. This interaction has been previously studied for degenerate levels and simple geometries involving infinite, perfectly conducting half-spaces or free space. Here, we generalize these studies by developing a convenient description that permits the analysis of these non-diagonal perturbations to the atomic Hamiltonian in terms of an accurate non-Hermitian Hamiltonian. Applying this theory to a hydrogen atom close to a dielectric nanoparticle, we show strong vacuum-field-induced state mixing that leads to drastic modifications in both the energies and decay rates compared to conventional diagonal perturbation theory. In particular, contrary to the expected Purcell enhancement, we find a surprising decrease of decay rates within a considerable range of atom-nanoparticle separations. Furthermore, we quantify the large degree of mixing of the unperturbed eigenstates due to the non-diagonal perturbation. Our work opens new quantum state manipulation possibilities in emitters with closely spaced energy levels.



# 1  Introduction

It is well known that atomic properties are modified due to the interaction with the quantized electromagnetic (EM) vacuum field supported by macroscopic bodies [1]. In the weak-coupling regime, this changes both the atomic linewidths (Purcell effect) [2] and energies (Lamb or Casimir-Polder [CP] shifts) [3]. These modifications have wide-ranging applications in fields such as optics or atomic and soft matter physics, including the design of efficient single photon sources [4–6], the atomic force microscope [7], new atom trapping methods [8,9] or the precise manipulation of atomic properties with tunable nanostructures [10]. Theoretical descriptions of these effects are commonly perturbative, using either standard perturbation theory or open quantum systems approaches [11], although efforts to go beyond the purely perturbative regime have also been published [12–14]. When the interactions are weak, the effect of the environment is customarily treated for each atomic state independently, giving rise to simple diagonal energy shifts and decay rates. However, for subsets of near-degenerate atomic states, the CP shift and/or spontaneous emission rates may be of the same order as the energy differences within the subset, suggesting that the above treatment is not consistent, even if the light-matter coupling is perturbative. This has been discussed in the literature for atoms in free space [15,16].

In this work, we show that the standard diagonal perturbation approach indeed fails when field-induced shifts are comparable to the energy level differences, requiring the treatment of environment-induced interactions between the levels [17,18]. Recently, this issue has been tackled with an open quantum systems' framework designed for structures with closely spaced levels [19]. In that work, the standard Bloch-Redfield equation [11] was turned into a Lindblad equation [20], with the corresponding benefit of guaranteed positive populations, while simultaneously eluding the usual secular approximation that neglects the couplings between non-degenerate states [21]. Here, we extend this framework to incorporate the effect of the counter-rotating terms in the light-matter Hamiltonian and construct a master equation that accurately represents the off-diagonal CP and decay terms, which we expect to be relevant in any system with subsets of near-degenerate levels. From the Lindblad equation, we extract an effective non-Hermitian Hamiltonian that determines the dynamics of a subset of levels and in turn enables a quantitative exploration of the vacuum-field-induced state mixing. We illustrate the effects of the off-diagonal terms by applying the above steps to a system comprised of a hydrogen atom close to an aluminum nitride (AlN) nanoparticle (NP), and study the impact of the off-diagonal couplings on the dynamics of the atom. We find strong modifications to the level structure and observe significant state mixing at atom-NP separations on the order of 100 nm. Consequently, the atomic dynamics close to the NP cannot be understood without consideration of the effects discussed in this work. The situation treated here lies between

conventional weak coupling (where light-matter interactions can be treated perturbatively and states can be considered independently) and strong coupling (where light and matter excitations mix significantly due to non-perturbative interactions). In this novel regime of "strong weak coupling", perturbative light-matter interactions lead to significant state mixing within the matter component.

## 2 Methods

### 2.1 Macroscopic quantum electrodynamics

We describe the interaction between atoms and the EM field supported by macroscopic bodies within macroscopic quantum electrodynamics (MQED) [22–24]. The corresponding Power-Zienau-Woolley light-matter Hamiltonian in the dipole approximation [3, 25–27] is

$$H = H_{\text{at}} + H_{\text{f}} - \mathbf{d} \cdot \mathbf{E}(\mathbf{r}_{\text{at}}). \tag{1}$$

Here, $H_{\text{at}}$ is the matter Hamiltonian, emphasizing that we treat a single atom. The field Hamiltonian,

$$H_{\text{f}} = \sum_{\lambda} \int \mathrm{d}^3 r \int \mathrm{d}\omega \, \hbar\omega \, \mathbf{f}_{\lambda}^{\dagger}(\mathbf{r}, \omega) \cdot \mathbf{f}_{\lambda}(\mathbf{r}, \omega), \tag{2}$$

contains the (bosonic) polaritonic annihilation and creation operators $\mathbf{f}_{\lambda}(\mathbf{r}, \omega)$ and $\mathbf{f}_{\lambda}^{\dagger}(\mathbf{r}, \omega)$ that describe both the purely electromagnetic and the macroscopic polarization fields. Here, the index $\lambda = \{e, m\}$ labels the electric or magnetic nature of the excitations, and the integrals are over all space and over all positive frequencies. The last term is the dipolar interaction between the atom with dipole operator $\mathbf{d}$ and the electric field $\mathbf{E}(\mathbf{r})$ evaluated at the atomic position $\mathbf{r}_{\text{at}}$, where

$$\mathbf{E}(\mathbf{r}) = \sum_{\lambda} \int \mathrm{d}^3 s \int \mathrm{d}\omega \, \mathbf{G}_{\lambda}(\mathbf{r}, \mathbf{s}, \omega) \cdot \mathbf{f}_{\lambda}(\mathbf{s}, \omega) + \text{H.c.}, \tag{3}$$

with

$$\mathbf{G}_e(\mathbf{r}, \mathbf{s}, \omega) = i \frac{\omega^2}{c^2} \sqrt{\frac{\hbar}{\pi\varepsilon_0} \operatorname{Im}\varepsilon(\mathbf{s}, \omega)} \mathbf{G}(\mathbf{r}, \mathbf{s}, \omega),$$

$$\mathbf{G}_m(\mathbf{r}, \mathbf{s}, \omega) = i \frac{\omega}{c} \sqrt{\frac{\hbar}{\pi\varepsilon_0} \frac{\operatorname{Im}\mu(\mathbf{s}, \omega)}{|\mu(\mathbf{s}, \omega)|^2}} \left[ \nabla_{\mathbf{s}} \times \mathbf{G}(\mathbf{s}, \mathbf{r}, \omega) \right]^T.$$

Here, $\varepsilon$ and $\mu$ stand for the electric and magnetic response functions, respectively. $\mathbf{G} = \mathbf{G}^0 + \mathbf{G}^{\text{scatt}}$ is the classical electromagnetic Green tensor, separated in its free-space and scattering contributions. In the weak-coupling regime, $\mathbf{G}^0$ is responsible for the free-space Lamb shift [3], a NP-independent contribution that can be simply reabsorbed in $H_{\text{at}}$. Compared to the effects we study in this work, this is a negligible correction that we discard entirely in the following.

For concreteness, we focus on a hydrogen atom interacting with a spheroidal AlN NP (see Figure 1a). It should be noted, however, that the following arguments are of broader generality and applicable to a wide range of physical systems, provided that the energies and transition dipole moments of the atom and the Green tensor of the nanostructure are accessible, a general requirement of CP calculations. We choose this NP shape and material for two reasons: (i) the EM resonances along the symmetry axis ($z$) enhance the atomic transitions mediated by $E_z$ with respect to the other components, and (ii) the energy range of the EM resonances coincides with the hydrogenic transition we want to target. Hence, this system provides a realistic and not

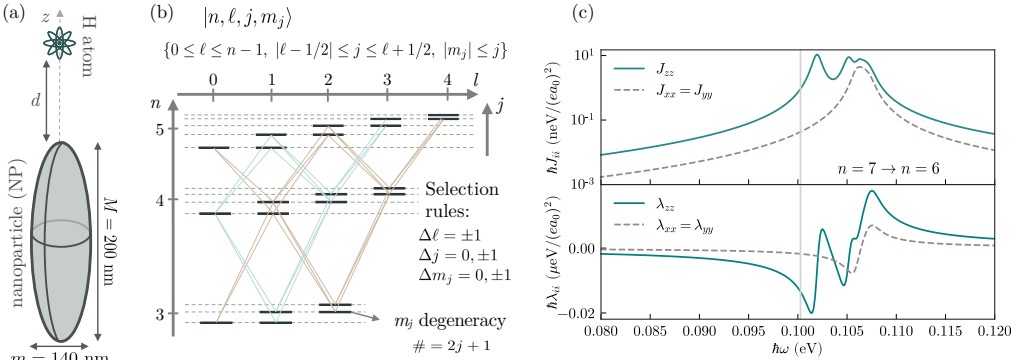

Figure 1: (a) Sketch of the system. (b) Simplified level structure of the hydrogen atom with Bohr levels and fine structure splitting (not to scale). Diagonal lines: dipolar transitions allowed from the $n = 4$ Bohr level to the $n = 5$ and $n = 3$ Bohr levels. (c) Top: spectral density of the AlN NP, obtained by setting $d = 50$ nm in Equation 7, bottom: result of the integral in Equation 6d. $e$ is the absolute value of the charge of the electron and $a_0$ is the Bohr radius. Vertical line: transition frequencies of the atom from $n = 7$ to $n = 6$.

overly complicated testing ground for the formalism developed, and will allow us to illustrate the effects of the off-diagonal vacuum shifts.

In Equation 1, $H_{\text{at}}$ is diagonal, and its eigenvalues include fine structure corrections [28]:

$$E_{nj} = \left( -\frac{1}{2n^2} - \frac{\alpha^2}{2n^3} \left[ \frac{1}{j + \frac{1}{2}} - \frac{3}{4n} \right] \right) E_{\text{h}}, \qquad (4)$$

where $n$, $j$, $\alpha$ and $E_{\text{h}} \simeq 27.2$ eV are the main quantum number, the total electronic angular momentum, the fine structure constant and the Hartree energy, respectively. The energy levels, schematically shown in Figure 1b, are distributed in well-separated Bohr levels labeled by $n$, corrected with the fine structure splitting $\Delta_F$, a $j$-dependent quantity that is 4 or more orders of magnitude smaller. This energy scale is small enough that the CP induced interaction between fine structure states with the same $n$ can be relevant. The NP is a spheroid with major and minor axes $M = 200$ nm and $m = 140$ nm, with the AlN dielectric permittivity taken from [29]. For this NP, the phonon-polariton resonances lie close to the transition energy between the hydrogenic $n = 7$ and $n = 6$ states. Specifically, we will focus on the off-diagonal effects within the $n = 7$ level for an atom located along the symmetry axis $z$ of the NP.

## 2.2 Master equation and effective non-Hermitian Hamiltonian

To describe the dynamics of the field-modified atomic levels and their mixing, we derive a Lindblad equation for the atomic density matrix $\rho$ by considering the EM fields as a weakly coupled bath and perturbatively tracing out the EM degrees of freedom. We start from the standard open quantum systems approach, which leads to the so-called Bloch-Redfield equa-

tion [11, 30]:

$$\dot{\rho} = -\frac{i}{\hbar}[H_{\text{at}}, \rho] + \sum_{abcd}\Bigg[ -i\Big(\Lambda_{ca,db}(\omega_{bd})|a\rangle\langle c|d\rangle\langle b|\rho - \Lambda_{ca,db}(\omega_{ac})\rho|a\rangle\langle c|d\rangle\langle b|\Big)$$
$$+ i\big[\Lambda_{ca,db}(\omega_{bd}) - \Lambda_{ca,db}(\omega_{ac})\big]|d\rangle\langle b|\rho|a\rangle\langle c|$$
$$- \frac{1}{2}\Big(\Gamma_{ca,db}(\omega_{bd})|a\rangle\langle c|d\rangle\langle b|\rho + \Gamma_{ca,db}(\omega_{ac})\rho|a\rangle\langle c|d\rangle\langle b|\Big)$$
$$+ \frac{1}{2}\big[\Gamma_{ca,db}(\omega_{bd}) + \Gamma_{ca,db}(\omega_{ac})\big]|d\rangle\langle b|\rho|a\rangle\langle c|\Bigg], \tag{5}$$

where $\rho$ is the atomic density matrix, the Latin indices $a, b, c, d$ denote atomic eigenstates and $\omega_{ab} = (E_a - E_b)/\hbar$. The rotationally invariant quantities $\Gamma_{ca,db}(\omega)$ and $\Lambda_{ca,db}(\omega)$ are given by

$$\Gamma_{ca,db}(\omega) = \mathbf{d}_{ca}^* \cdot \boldsymbol{\gamma}(\omega) \cdot \mathbf{d}_{db}, \tag{6a}$$
$$\Lambda_{ca,db}(\omega) = \mathbf{d}_{ca}^* \cdot \boldsymbol{\lambda}(\omega) \cdot \mathbf{d}_{db}, \tag{6b}$$

where $\mathbf{d}_{ca}$ is a matrix element of the atomic dipole operator, and $\boldsymbol{\gamma}$ and $\boldsymbol{\lambda}$ are defined as

$$\boldsymbol{\gamma}(\omega) = 2\pi \mathbf{J}(\omega), \tag{6c}$$
$$\boldsymbol{\lambda}(\omega) = \mathcal{P}\int d\omega' \frac{\mathbf{J}^{\text{scatt}}(\omega')}{\omega - \omega'}. \tag{6d}$$

Here, $\mathcal{P}$ denotes the principal value and $\mathbf{J}(\omega)$ is the spectral density of the EM field,

$$\mathbf{J}(\omega) = \frac{\omega^2}{\hbar\pi\epsilon_0 c^2}\,\text{Im}\,\mathbf{G}(\mathbf{r}_{\text{at}}, \mathbf{r}_{\text{at}}, \omega). \tag{7}$$

We have replaced $\mathbf{G}$ by $\mathbf{G}^{\text{scatt}}$ in $\mathbf{J}^{\text{scatt}}$ (Equation 6d), since the free-space contribution is assumed to be included in the bare atomic Hamiltonian, and it is also smaller than the NP-induced effects discussed in this work. We note that while we indicate the complex conjugation of the dipole matrix elements, it is possible to choose an atomic basis in which they are real; thus, both $\Gamma$ and $\Lambda$ would be real quantities. Note that the expressions for $\boldsymbol{\lambda}$ contain both the so-called resonant contributions, which can be shown to be proportional to $\text{Re}\,\mathbf{G}^{\text{scatt}}$, and non-resonant contributions to the energy shift [31]. The electromagnetic Green tensor is computed using the boundary element method implemented in SCUFF-EM [32, 33].

Equation 5 does indeed describe the bath-induced interaction between levels, but allows for non-physical dynamics, as it does not guarantee the positivity of the density matrix. A standard secularization procedure leads to a completely positive Lindblad equation, but removes the crucial off-diagonal terms describing state mixing (for more details see subsection A.2 of the appendix). Instead of secularization, we extend the approach of Ref. [19] to obtain a completely positive Lindblad equation for near-degenerate levels by including the effect of the counter-rotating terms of the dipolar interaction. This approach consists in replacing both $\Lambda_{ca,db}(\omega_{bd})$ and $\Lambda_{ca,db}(\omega_{ac})$ with their geometric mean $\tilde{\Lambda}_{ca,db} = \sqrt{\Lambda_{ca,db}(\omega_{bd})}\sqrt{\Lambda_{ca,db}(\omega_{ac})}$, and the same for $\Gamma_{ca,db}(\omega_{bd})$ and $\Gamma_{ca,db}(\omega_{ac})$. Similar ideas have also been proposed elsewhere in the literature [18, 34, 35]. When applied to Equation 5, this replacement symmetrizes the pairs of indices $ca$ and $db$. Then, for the symmetric geometry of Figure 1a where $\mathbf{G}$ is a diagonal matrix, the resulting master equation can be rewritten as

$$\dot{\rho} = \frac{-i}{\hbar}[H_{\text{at}} + H_{\text{CP}}, \rho] + \sum_{\delta, n} L_{\Sigma_\delta^{(n)}}[\rho]. \tag{8}$$

Here, $H_{\text{CP}}$ is the CP shift, $\Sigma_\delta^{(n)}$ are decay operators, and $L_A[\rho] = A\rho A^\dagger - \frac{1}{2}\{A^\dagger A, \rho\}$ is a Lindblad dissipator. There is a decay operator for each spatial component in the spherical basis $\delta = 0, \pm 1$, and for each Bohr level $n$. The CP shift is given by $H_{\text{CP}} = \hbar \sum_{\delta,n} D_\delta^{(n)\dagger} D_\delta^{(n)}$, where

$$\langle j|D_\delta^{(n)}|n\rangle = \sqrt{\lambda_{\delta\delta}(\omega_{nj})}d_{jn}^\delta, \tag{9a}$$

and the decay operators can be expressed as

$$\langle j|\Sigma_\delta^{(n)}|n\rangle = \sqrt{\gamma_{\delta\delta}(\omega_{nj})}d_{jn}^\delta. \tag{9b}$$

In the definitions above, the atomic states $|n\rangle$ and $|j\rangle$ belong to the $n$th and $j$th Bohr level, respectively. Had the counter-rotating terms in the dipolar coupling not been taken into account, the Hermitian dipole operators in Equation 5 would have been replaced by raising and lowering operators, such that only terms where $\omega_j \leq \omega_n$ would be present in Equation 8. Instead, our description also incorporates the CP shift contribution given by states with $\omega_j > \omega_n$. More details about the derivation steps can be found in subsection A.3 of the appendix.

There are two details that require addressing to completely justify the above Lindblad master equation. The first one relates to the free-space Lamb shift. Although we have done the previous manipulations already assuming that $\mathbf{G}^{\text{scatt}}$ is the only relevant part of the energy shift (see Equation 6d), strictly speaking there is also the free-space part. Even if the free-space Lamb shift is indeed negligible compared to the NP-assisted shift, the geometric mean introduces a cross-term between free-space and scattered contributions that one could expect to be larger than the pure free-space shift. However, if $\lambda_i = \lambda_{i,0} + \lambda_{i,s}$ then the shift can be expanded as $\sqrt{\lambda_1\lambda_2} \approx \bar{\lambda}_s + \frac{1}{2}\left(\frac{\bar{\lambda}_s}{\lambda_{1,s}}\lambda_{1,0} + \frac{\bar{\lambda}_s}{\lambda_{2,s}}\lambda_{2,0}\right)$, where $\bar{\lambda}_s = \sqrt{\lambda_{1,s}\lambda_{2,s}}$. Since the dynamically relevant terms are those where $\lambda_{1,s} \approx \lambda_{2,s} \approx \bar{\lambda}_s$, the final correction due to the free-space contribution turns out to be $\approx \frac{1}{2}(\lambda_{1,0} + \lambda_{2,0})$, that is, of the same order as the free-space shift itself. Therefore, the simple argument to discard the free-space contribution turns out to be enough for the cross-terms that appear with the geometric mean too. Finally, we note that for an atom interacting with a thermal bath, the expression in Equation 8 would contain additional terms proportional to the occupation number $n_T(\omega) = (e^{\hbar\omega/k_B T} - 1)^{-1}$, but at the relevant transition frequencies this can be shown to be very small for laboratory temperatures. Therefore, these thermal contributions have been neglected.

While solving Equation 8 is considerably more affordable than a direct solution of the Schrödinger equation with the Hamiltonian from Equation 1, several faithful approximations allow for further simplification, succinctly described below, with more details and an explicit check of the validity of the approximations given in Appendix B and Appendix C, respectively. First, for the dynamics within a single Bohr level, here $n = 7$, we can discard the states with $n \neq 7$ and write a closed set of equations for $n = 7$, due to the large difference in the energy scales associated to the Bohr transition energies and the environment-induced perturbations. The other states are then considered only implicitly as intermediate virtual states that contribute to the CP and decay terms. Furthermore, within this subspace, the "quantum jump" terms $\Sigma_\delta^{(7)}\rho\Sigma_\delta^{(7)\dagger}$ in Equation 8 are negligible since they are proportional to $J_{\delta\delta}(\Delta_F)$, and the spectral density approaches zero for small frequencies. With these approximations, the dynamics within the $n = 7$ subspace is described by an effective non-Hermitian Hamiltonian

$$H_{\text{eff}}^{(7)} = H_{\text{at}}^{(7)} + \hbar \sum_\delta \left(D_\delta^{(7)\dagger}D_\delta^{(7)} - \frac{i}{2}\Sigma_\delta^{(7)\dagger}\Sigma_\delta^{(7)}\right), \tag{10}$$

where $H_{\text{at}}$ has been projected onto the $n = 7$ subspace. Last, due to the axial symmetry of the system (see Figure 1a), the $z$ component of the total angular momentum is conserved and Equation 10 consists of independent blocks for each value of $m_j$. In the following, we focus on the subspace $m_j = 1/2$, which reduces the number of states to be considered to 7 for this particular case.

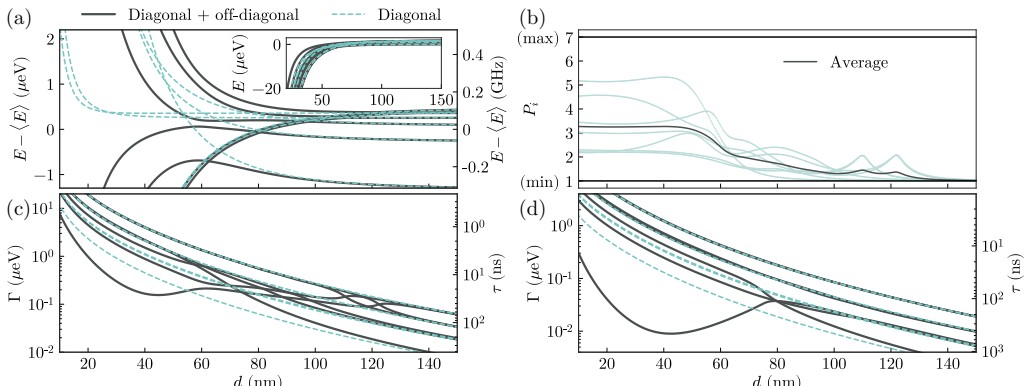

Figure 2: Every quantity is plotted against the atom-NP separation $d$. (a) Energies of the eigenstates of the effective non-Hermitian Hamiltonian. (b) Participation ratios $P_i$ indicating the degree of mixing of the eigenstates. (c) Decay rates of the eigenstates of the effective non-Hermitian Hamiltonian. (d) Decay rates with the NP shape chosen to optimize the decay rate reduction: $M = 200$ nm and $m = 120$ nm. For plots (a), (c) and (d), the solid black lines correspond to the full model with off-diagonal terms, and the green dashed lines to the model without the off-diagonal terms. For plot (b), the green lines are the participation ratio of each eigenstate, and the black line represents the average participation ratio.

## 3 Results

The above derivation significantly simplifies the analysis of the dynamics. In particular, the effective Hamiltonian can be diagonalized, and the real and imaginary parts of its eigenvalues correspond to the energies and decay rates of the states including the vacuum-field-induced state mixing. These energies and decay rates are shown in Figure 2a and Figure 2c, respectively. Since the CP shift is dominated by an overall attraction to the surface (inset of Figure 2a), we plot it relative to the average value for each separation $d$, revealing a completely different and much more complex structure compared to the fully secularized, diagonal model. In particular, clear avoided crossings highlight the relevance of the off-diagonal terms, similar to effects found in interatomic interactions in [36], but here occurring within a single atom. Similarly striking differences between both models appear in the decay rates shown in Figure 2c. Due to the off-diagonal terms, the decay rates cross each other several times. Particularly prominent is the vacuum-field-induced generation of a state that becomes more protected against spontaneous decay as the atom approaches the NP for separations between about $d = 40$ nm and 60 nm. This is in stark contrast to the behavior when the states are treated independently, for which the effect of quenching leads to monotonic increase of the Purcell factor and thus decay rate with decreasing separation [37, 38]. Here, the same environment that induces the decay also produces the interactions that mix the states and leads to the formation of a state protected from the influence of the environment. We note that the subradiant or metastable state created by field-induced mixing has a smaller decay rate than any of the original eigenstates of the atom at the same distance when mixing is not included.

The emergence of this protected state can be understood by realizing that the system approaches an idealized situation in which one of the states in the $n = 7$, $m_j = 1/2$ manifold is fully decoupled from the EM environment. This situation would occur if we could ignore the (i) fine structure, (ii) $x$- and $y$-polarized electric fields in Equation 1, and (iii) contributions of states outside of the $n = 6$ Bohr level to the CP shift and spontaneous decay. Then, 7 states in $n = 7$ couple to 6 states in $n = 6$ through the single operator $d_z$, and there is always one

superposition ("dark state") with vanishing coupling. For the realistic system, this idealized situation is approached for various reasons. First, the elongated shape of the NP suppresses the $xx$ and $yy$ components of $J$ and $\lambda$ compared to the $zz$ component. Second, the coincidence of the first peak of $J_{zz}$ and $\lambda_{zz}$ with the energy of the transition from $n = 7$ to $n = 6$ enhances the contributions from $n = 6$ intermediate states compared to other Bohr levels. Lastly, when the CP shifts become greater than the fine structure, the latter becomes a perturbative correction that can be neglected to lowest order. Based on these considerations, we change the aspect ratio of the NP by decreasing the minor axis to $m = 120$ nm in order to amplify the protection of the state. As shown in Figure 2d, the minimum decay rate becomes an order of magnitude smaller than the naive expectation without off-diagonal terms, unambiguously demonstrating that the off-diagonal terms can significantly impact the structure of the atom and cannot be neglected in a realistic description.

Next, we evaluate the amount of vacuum-field-induced state mixing. The eigenstates $|\psi\rangle$ of Equation 10 are linear superpositions of the fine structure basis states $|\phi_k\rangle$, and the degree of this mixing can be quantified using the so-called participation ratio $P$ [39], defined as

$$P(|\psi\rangle) = \left[ \sum_k |\langle \psi | \phi_k \rangle|^4 \right]^{-1} . \tag{11}$$

It measures the number of basis states "equally" contributing to the normalized state $|\psi\rangle$, with possible values ranging from 1 to the number of basis states (7 for the case studied here). For example, for a state of the form $|\psi\rangle = \sqrt{1/n} \sum_{k=1}^{n} e^{i\theta_k} |\phi_k\rangle$, $P$ equals $n$. In Figure 2b, we show the participation ratio of the eigenstates of Equation 10 as a function of the atom-NP separation $d$, with green lines showing $P$ for each eigenstate and the thick black line representing the average over all states. We find state mixing to be negligible for separations above $\approx 150$ nm, indicating that the off-diagonal contributions to Equation 10 are too small to effectively couple the states. For shorter distances, the magnitude of the off-diagonal terms, $\left| \langle i | H_{\text{eff}} | j \rangle \right|$, becomes comparable to the difference of the corresponding diagonal elements, $\left| \langle i | H_{\text{eff}} | i \rangle - \langle j | H_{\text{eff}} | j \rangle \right|$, and the states mix appreciably. In particular, clear peaks in $P$ appear when the diagonal CP shifts bring initially detuned states into resonance, such that the off-diagonal elements dominate more easily. Despite the non-monotonic behavior, overall the participation ratios tend to increase as the atom approaches the NP until they stabilize at about $d = 30$ nm. At closer distances, vacuum-field couplings determine the eigenstates and dominate over the fine structure, which becomes a small perturbation of these new eigenstates. We note that for the setup studied here, almost complete mixing of some atomic states is achieved, with values of $P$ larger than 5, close to the theoretical maximum of 7.

## 4 Conclusion

In conclusion, we have shown that vacuum-field-induced interactions can significantly mix groups of near-degenerate levels in atoms and must, therefore, be included to accurately characterize the dynamics. We have derived a completely positive Lindblad master equation that provides a precise description of these situations and allows an interpretation of the field-modified atomic structure in terms of an effective non-Hermitian Hamiltonian. For concreteness, we have applied these general ideas to a hydrogen atom coupled to an AlN NP. This leads to striking new features in the atomic structure, such as avoided level crossings and, surprisingly, a decrease of the decay rate for a particular eigenstate with decreasing distance to the NP even though the Purcell factor for each uncoupled state grows monotonically. This illustrates that the off-diagonal terms can even have counterintuitive consequences. Deeper exploration of the eigenstates reveals that the atomic structure in this regime greatly differs

from the original fine structure of the free-space atom, even though the atom-field interactions are perturbative and the atom remains a well-defined entity, in contrast to the strong light-matter coupling regime. From an atomic physics' perspective, the hydrogen atom treated here becomes "unrecognizable" as the atomic structure and spectroscopic properties within each sublevel change completely. We note that while we treat a specific setup here, the framework can be straightforwardly applied to other nanostructures (e.g., graphene [10]) and emitters or level splittings (e.g., due to hyperfine structure in Rydberg atoms [36,40]). Our work thus extends the regime where vacuum-field-induced forces and decay rates are accurately described and opens the door to new strategies for developing quantum state manipulation platforms based on off-diagonal vacuum-induced effects.

# 5 Acknowledgements

**Funding information** This work has been funded by the Spanish Ministry of Science, Innovation and Universities-Agencia Estatal de Investigación through the FPI contract No. PRE2019-090589 as well as grants RTI2018-099737-BI00, PID2021-125894NB-I00, and CEX2018-000805-M (through the María de Maeztu program for Units of Excellence in R&D). We also acknowledge financial support from the Proyecto Sinérgico CAM 2020 Y2020/TCS-6545 (NanoQuCo-CM) of the Community of Madrid, and from the European Research Council through grant ERC-2016-StG-714870.

# A Derivation of the master equation

The derivation of the Lindblad master equation used in this work, Equation 8, is closely based on the one presented in [19], modified to include the counter-rotating terms of the light-matter Hamiltonian and a realistic electromagnetic environment, with three spatial components and non-trivial structure. We revisit the derivation here and highlight the additions and differences compared to [19]. For simplicity, the derivation is presented in a way that directly relates to the illustrative physical system of the main text, that is, a hydrogen atom. However, this is not a true limitation of the approach and approximations, as long as one considers level structures with distinct subsets of closely spaced states, a common feature of atomic systems due to fine structure or hyperfine structure splittings. We start by describing features of the Bloch-Redfield (BR) equation, and then explain the customary secular approximation, a procedure known to yield a completely positive Lindblad master equation. This equation would systematically neglect the off-diagonal terms discussed in this work. Then, we take the BR equation and perform a series of approximations that lead to a different Lindblad equation, maintaining the non-secular terms.

## A.1 Comments on the Bloch-Redfield master equation

The BR equation for our system is given by Equation 5. Although it is rather complicated, the physical interpretation of each line is simple: the first two lines of the sum are responsible for the Casimir-Polder (CP) shifts, while the third and fourth lines describe decay processes. While the CP terms can often be neglected, this is not the case for the system we study, as they are of the same order as the hydrogenic fine structure. For the same reason, we must include the counter-rotating (CR) terms in the full light-matter Hamiltonian. Otherwise, $\lambda(\omega)$ would only be evaluated at non-negative frequencies and Equation 5 would miss significant contributions to the CP terms arising from the negative frequencies. The CR terms only affect

the energy shift, as the decay terms $\gamma(\omega)$ vanish at negative frequencies. It is worth noting that even without considering the CR terms, the equation already includes the basis for the off-diagonal CP terms we discuss in the main text, albeit in a complex manner that is hard to disentangle.

The BR equation has several drawbacks: First, it does not guarantee positivity of the density matrix. Although it has been shown that these deviations from physical density matrices are negligible when the approximations made in deriving the BR equation are valid [21, 41], dealing with formally unphysical density matrices requires additional care. Second, the BR matrix is characterized by a superoperator of dimension $N^2 \times N^2$, where $N$ is the number of system states, which makes analysis of its behavior challenging. In contrast, a Lindblad-type master equation automatically ensures the physicality of the density matrix, and at the same time allows for a simpler analysis since it is characterized by a single Hamiltonian and a set of decay operators, all of dimensions $N \times N$.

## A.2 Lindblad master equation with full secularization

The usual procedure to obtain a Lindblad equation from Equation 5 is the so-called secular approximation which consists in eliminating every term where $\omega_{ac} \neq \omega_{bd}$. Doing so yields

$$
\dot{\rho} = -\frac{i}{\hbar}[H_{\mathrm{at}}, \rho] - i \sum_{abd}^{(S)} \Big[ \Lambda_{da,db}(\omega_{bd})|a\rangle\langle b|, \rho \Big]
$$
$$
+ \sum_{abcd}^{(S)} \Gamma_{ca,db}(\omega_{bd}) \bigg( |d\rangle\langle b|\rho|a\rangle\langle c| - \frac{1}{2}\big\{ |a\rangle\langle c|d\rangle\langle b|, \rho \big\} \bigg), \tag{A.1}
$$

where the superscript $(S)$ in the sum indicates that only terms with $\omega_{ac} = \omega_{bd}$ are kept. In the energy shift, this is equivalent to the condition $\omega_a = \omega_b$ since $|c\rangle = |d\rangle$ there. The energy shift is clearly Hermitian because it is a real and symmetric matrix. The decay term can be reexpressed by grouping the sum over transitions into sets with a given frequency $\Omega = \omega_{ac} = \omega_{bd}$, which yields

$$
\sum_{\Omega}\sum_{\alpha\beta} \gamma_{\alpha\beta}(\Omega)\bigg( \sigma_\Omega^\beta \rho \sigma_\Omega^{\alpha\dagger} - \frac{1}{2}\big\{ \sigma_\Omega^{\alpha\dagger}\sigma_\Omega^\beta, \rho \big\} \bigg) = \sum_{\Omega\epsilon} \Gamma_\epsilon(\Omega)\bigg( S_\Omega^\epsilon \rho S_\Omega^{\epsilon\dagger} - \frac{1}{2}\big\{ S_\Omega^{\epsilon\dagger}S_\Omega^\epsilon, \rho \big\} \bigg). \tag{A.2}
$$

Here, Greek indices $\alpha, \beta$ indicate spatial directions, while all dipole transitions $d^\alpha$ with a frequency difference of $\Omega$ are combined in the transition operators $\sigma_\Omega^\alpha = \sum_{ab}^{(\Omega)} d_{ab}^\alpha |a\rangle\langle b|$. The right-hand side above is obtained by diagonalizing the positive definite matrix $\gamma_{\alpha\beta}(\Omega) = \sum_\epsilon M_{\alpha\epsilon}^\dagger(\Omega)\Gamma_\epsilon(\Omega)M_{\epsilon\beta}(\Omega)$ for each transition frequency $\Omega$ and defining $S_\Omega^\epsilon = \sum_\alpha M_{\epsilon\alpha}(\Omega)\sigma_\Omega^\alpha$. In this last form, it is evident that the full secularization returns a Lindblad master equation. However, the only off-diagonal terms present are the ones connecting degenerate states. This approximation has been shown to be inadequate in a variety of contexts [19, 21, 41], since it indiscriminately removes the coupling between coherences (off-diagonal elements of $\rho$) and populations of non-degenerate states. Thus, in the system explored in the main text, relevant physics would be omitted within each Bohr level.

## A.3 Lindblad master equation: Derivation details and proof

We here show how to derive the Lindblad equation including off-diagonal terms between non-degenerate states used in the main text from the BR equation, Equation 5. Instead of a full secularization as discussed above, we start by performing a partial secularization to discard terms where the timescale induced by the environment, $\tau_E \sim \min(|\mathbf{d}^2\lambda|^{-1}, |\mathbf{d}^2\gamma|^{-1})$, is much larger than that of the atomic transitions, $\tau_{\mathrm{at}} \sim |\omega_{ac} - \omega_{bd}|^{-1}$. This is not an important

step, but it significantly simplifies the resulting expressions. In Equation 5, this is fulfilled for terms where $|a\rangle$ and $|b\rangle$ belong to different Bohr levels: First, if either $|c\rangle$ or $|d\rangle$ belongs to a different Bohr level than $|a\rangle$ and $|b\rangle$, respectively, then $\tau_{\mathrm{at}}$ is very small compared with $\tau_E$, and secularization is well-justified. If, instead, $|c\rangle$ and $|d\rangle$ belong to the same Bohr levels as $|a\rangle$ and $|b\rangle$, respectively, then $\tau_E \propto 1/\gamma(\Delta_F/\hbar)$ becomes extremely large because $\Delta_F$ is on the scale of the fine structure splitting and the spectral density approaches 0 when $\omega$ goes to 0. Hence, even if $\tau_{\mathrm{at}} \sim \hbar/\Delta_F$ is large, $\tau_E$ is even larger in the system studied here. This partially secularized BR equation is significantly simpler than the full one, but is not yet in Lindblad form.

We next apply the geometric mean replacement discussed in the main text to Equation 5 and obtain

$$\dot{\rho} = -\frac{i}{\hbar}[H_{\mathrm{at}}, \rho] - i\left[\sum_{abcd}^{(s)} \tilde{\Lambda}_{ca,db} |a\rangle\langle c|d\rangle\langle b|, \rho\right] + \sum_{abcd}^{(s)} \tilde{\Gamma}_{ca,db}\left(|d\rangle\langle b|\rho|c\rangle\langle a| - \frac{1}{2}\{|a\rangle\langle c|d\rangle\langle b|, \rho\}\right), \quad \text{(A.3)}$$

where the superscript $(s)$ in the sum indicates the partial secularization mentioned above. This procedure is accurate for the following reason. When the spectral density, given in Equation 7, is slowly varying: $J(\omega + |\mathbf{d}^2\lambda|) \simeq J(\omega)$ and $J(\omega + |\mathbf{d}^2\gamma|) \simeq J(\omega)$. In that case, for each term in Equation 5 where $|\omega_{ac} - \omega_{bd}| < \max(|\mathbf{d}^2\gamma|, |\mathbf{d}^2\lambda|)$, the change in the value of the element is small, and the geometric mean is a good approximation. For terms where $|\omega_{ac} - \omega_{bd}| > \max(|\mathbf{d}^2\gamma|, |\mathbf{d}^2\lambda|)$, the value might change appreciably, but its effect on the dynamics is small due to the difference in energy scales. In fact, such terms could be eliminated through an additional secularization to a good approximation.

After the replacement, the CP Hamiltonian, $H_{\mathrm{CP}} = \hbar \sum_{abc}^{(s)} \tilde{\Lambda}_{ca,cb} |a\rangle\langle b|$, contains both diagonal and off-diagonal matrix elements. We note that the effect of the CR terms of the light-matter coupling is manifested in the precise values of the matrix elements, which change significantly depending on whether the CR interactions are included or not. Careful analysis shows that this Hamiltonian is Hermitian if $\lambda(\omega_{ca})$ and $\lambda(\omega_{bd})$ have the same sign. Since $\lambda(\omega)$ has sign changes, in general, this is not always necessarily satisfied, leading to a potentially problematic, non-Hermitian CP Hamiltonian. In the cases studied in the manuscript, the partial secularization we performed earlier ensures that only terms with $\omega_{ac} \simeq \omega_{bc}$ survive, and combined with the slow-varying property of the spectral density, the sign condition is satisfied always. A discussion of the general situation where this is not necessarily true will be presented in a future work.

In order for Equation A.3 to be a Lindblad-type master equation, the decay rate tensor $\tilde{\Gamma}_{ca,db}$ interpreted as a matrix in the combined indices $ca$ and $db$, sometimes called Kossakovski matrix, has to be positive semidefinite. Then, it can be diagonalized with non-negative eigenvalues and the last term in Equation A.3 can be rewritten as a sum of standard Lindblad decay terms. While it is symmetric by construction, we are not aware of a general proof of positive semidefiniteness of the decay tensor that results when the procedure described above is applied to arbitrary spectral densities and atomic spectra. For the cases we treat in the manuscript, where the Green tensor is diagonal and cylindrically symmetric such that its Cartesian components satisfy $G_{xx} = G_{yy}$, we give a proof below through explicit construction of the diagonalized form. Under this assumption, the decay tensor has the form $\gamma(\omega) = \mathrm{diag}(\gamma_{xx}(\omega), \gamma_{xx}(\omega), \gamma_{zz}(\omega))$. We now express $\Gamma_{ca,db}(\omega)$ in terms of the spherical basis defined by

$$\mathbf{d}' = \begin{bmatrix} d^{+1} \\ d^{-1} \\ d^0 \end{bmatrix} = \mathbf{U} \cdot \mathbf{d} = \begin{bmatrix} -1/\sqrt{2} & -i/\sqrt{2} & 0 \\ 1/\sqrt{2} & -i/\sqrt{2} & 0 \\ 0 & 0 & 1 \end{bmatrix} \cdot \begin{bmatrix} d^x \\ d^y \\ d^z \end{bmatrix}. \quad \text{(A.4)}$$

By construction, the spherical components of the dipole operator, $d^\delta$, connect states with a given $m_j$ to states with $m_j + \delta$. Due to its symmetry, $\boldsymbol{\gamma}(\omega)$ is invariant under transformation to the spherical basis, $\boldsymbol{\gamma}'(\omega) = \mathbf{U} \cdot \boldsymbol{\gamma}(\omega) \cdot \mathbf{U}^\dagger = \boldsymbol{\gamma}(\omega)$. Since $m_j$ is a well-defined quantum number of our basis states, the advantage of the spherical basis is that every transition operator $|a\rangle\langle c|$ allowed by the selection rules (see Fig. 1b of the main text) is mediated by only one of $d^{+1}$, $d^{-1}$ or $d^0$. Furthermore, because of the diagonal form of $\boldsymbol{\gamma}'$, the transition operators $|d\rangle\langle b|$ and $|c\rangle\langle a|$ must have the same $\delta$; otherwise $\Gamma_{ca,db}(\omega) = 0$. As a consequence, we can expand the last term of Equation A.3 as three separate sums, one for each value of $\delta$, indicated below with the label $\delta$ on the second summation sign:

$$
\sum_\delta \sum_{abcd}^{(s,\delta)} \sqrt{\Gamma_{ac,db}(\omega_{bd})} \sqrt{\Gamma_{ac,db}(\omega_{ac})} \left[ |d\rangle\langle b|\rho|a\rangle\langle c| - \frac{1}{2}\left\{ |a\rangle\langle c|d\rangle\langle b|, \rho \right\} \right]
$$
$$
= \sum_\delta \sum_{abcd}^{(s,\delta)} d_{ca}^{\delta*} d_{db}^\delta \sqrt{\gamma_{\delta\delta}(\omega_{bd})} \sqrt{\gamma_{\delta\delta}(\omega_{ac})} \left[ |d\rangle\langle b|\rho|a\rangle\langle c| - \frac{1}{2}\left\{ |a\rangle\langle c|d\rangle\langle b|, \rho \right\} \right]
$$
$$
= \sum_{\delta n} \left( \Sigma_\delta^{(n)} \rho \Sigma_\delta^{(n)\dagger} - \frac{1}{2}\left\{ \Sigma_\delta^{(n)\dagger} \Sigma_\delta^{(n)}, \rho \right\} \right), \tag{A.5}
$$

where we have used that $\gamma_{\delta\delta}(\omega) > 0$ and that the $d_{ca}^\delta$ are real for any pair $ca$, and in the last step, we have defined the summed transition operator

$$
\Sigma_\delta^{(n)} = \sum_{db}^{(n)} d_{db}^\delta \sqrt{\gamma_{\delta\delta}(\omega_{bd})} |d\rangle\langle b|. \tag{A.6a}
$$

Here, the states $|b\rangle$ belong to the same Bohr level with main quantum number equal to $n$, while the $|d\rangle$ states can belong to any Bohr level. This simplification is a consequence of the partial secularization explained at the beginning of this subsection. In this form, the decay term is given by an explicit Lindblad operator in terms of just three decay operators for each Bohr level $n$. We note that identical manipulations can be done on the energy shift terms, which can be refactored as

$$
D_\delta^{(n)} = \sum_{db}^{(n)} d_{db}^\delta \sqrt{\lambda_{\delta\delta}(\omega_{bd})} |d\rangle\langle b|. \tag{A.6b}
$$

Finally, we can rewrite Equation A.3 in our system as

$$
\dot\rho = -\frac{i}{\hbar} \left[ H_{\text{at}} + H_{\text{CP}}, \rho \right] + \sum_{\delta n} L_{\Sigma_\delta^{(n)}}[\rho], \tag{A.7}
$$

where $H_{\text{CP}} = \hbar \sum_{\delta n} D_\delta^{(n)\dagger} D_\delta^{(n)}$ and $L_A[\rho] = A\rho A^\dagger - \frac{1}{2}\{A^\dagger A, \rho\}$ is a standard Lindblad decay term. This is indeed a Lindblad equation for the atom that includes the relevant off-diagonal couplings both in the CP shift and the decay term.

## B Derivation of the effective Hamiltonian

Any Lindblad equation $\dot\rho = -\frac{i}{\hbar}[H, \rho] + \sum_j L_{A_j}[\rho]$ can be rewritten as $\dot\rho = -\frac{i}{\hbar}\left(H_{\text{eff}}\rho - \rho H_{\text{eff}}^\dagger\right) + \sum_j A_j \rho A_j^\dagger$, with the effective non-Hermitian Hamiltonian $H_{\text{eff}} = H - \frac{i}{2}\sum_j A_j^\dagger A_j$, and the terms of the last sum commonly referred to as the "refilling" or "quantum jump" terms. In physical situations where the refilling terms are negligible, the

dynamics are then fully characterized by the eigenstates and eigenvalues of the effective Hamiltonian [42]. In the main text, we are concerned with the dynamics within a given Bohr level, in particular $n = 7$. Due to the partial secularization we performed, the effective Hamiltonian associated with Equation A.7 is block-diagonal in Bohr levels, such that $n$ remains a good quantum number and $[H_{\text{eff}}, \mathcal{P}_n] = 0$, where $\mathcal{P}_n$ is a projection operator onto the subspace with principal quantum number $n$. Projecting the Lindblad master equation onto this subspace gives

$$\dot{\rho}_n = -\frac{i}{\hbar}\left(H_{\text{eff}}^{(n)}\rho_n - \rho_n H_{\text{eff}}^{(n)\dagger}\right) + \sum_{\delta n'} \mathcal{P}_n \Sigma_\delta^{(n')} \rho \Sigma_\delta^{(n')\dagger} \mathcal{P}_n \, , \tag{B.1}$$

$$H_{\text{eff}}^{(n)} = \mathcal{P}_n H_{\text{eff}} \mathcal{P}_n = H_{\text{at}}^{(n)} + \hbar \sum_\delta \left(D_\delta^{(n)\dagger} D_\delta^{(n)} - \frac{i}{2}\Sigma_\delta^{(n)\dagger}\Sigma_\delta^{(n)}\right) \, , \tag{B.2}$$

where $\rho_n = \mathcal{P}_n \rho \mathcal{P}_n$ and $H_{\text{at}}^{(n)} = \mathcal{P}_n H_{\text{at}} \mathcal{P}_n$. We thus only need to show that the refilling terms are negligible for the dynamics within a given Bohr level. To this end, we can rewrite them as

$$\mathcal{P}_n \Sigma_\delta^{(n')} \rho \Sigma_\delta^{(n')\dagger} \mathcal{P}_n = \sum_{abcd}^{(s,\delta)} d_{ca}^{\delta *} d_{db}^{\delta} \sqrt{\gamma_{\delta\delta}(\omega_{bd})}\sqrt{\gamma_{\delta\delta}(\omega_{ac})}\mathcal{P}_n|d\rangle\langle b|\rho|a\rangle\langle c|\mathcal{P}_n \, . \tag{B.3}$$

Because of the partial secular approximation, $|a\rangle$ and $|b\rangle$ belong to the same Bohr level $n'$, and due to the projection operators $\mathcal{P}_n$, $|c\rangle$ and $|d\rangle$ also have the same principal quantum number, $n$. Also, because $\gamma$ is only non-zero at positive frequencies, we have that $n' \geq n$. We can immediately discard terms with $n' > n$: They refer to the population that flows into the Bohr level $n$ through spontaneous emission from higher-lying Bohr levels, but since we assume that the initial atomic state is in level $n$ and there are no processes leading to higher levels, these terms do not contribute. For the remaining terms with $n' = n$, the atomic time scales are $\tau_{\text{at}} \sim \hbar/\Delta_F$, but the decay-induced time scales are $\tau_E \propto 1/\gamma(\Delta_F/\hbar)$. Given the spectral density used in the main text, in our system $\tau_E \gg \tau_{\text{at}}$. Thus, the effect of the terms with $n' = n$ is negligible, and we can safely remove the "refilling" term and write the dynamics in the subspace with principal quantum number $n$ as

$$\dot{\rho}_n = -\frac{i}{\hbar}\left(H_{\text{eff}}^{(n)}\rho_n - \rho_n H_{\text{eff}}^{(n)\dagger}\right) \, , \tag{B.4}$$

which is equivalent to the Schrödinger equation $\partial_t |\psi(t)\rangle = -\frac{i}{\hbar} H_{\text{eff}}^{(n)} |\psi(t)\rangle$.

## B.1 Angular momentum conservation

For each Bohr level $n$, the effective Hamiltonian Equation B.2 derived above is a block diagonal matrix, with each block corresponding to a given value of the $z$-projection $m_j$ of the atomic angular momentum. This is easy to see since $H_{\text{at}}$ conserves angular momentum, while the operators $D_\delta^{(n)}$ and $\Sigma_\delta^{(n)}$ connect $m_j$ to $m_j + \delta$, and their Hermitian conjugates connect $m_j + \delta$ back to $m_j$, such that overall, $m_j$ is conserved. In contrast, physically and in the full Lindblad master equation Equation A.7, it is only the $z$-projection of the total angular momentum of the photons and atom together that is conserved due to the cylindrical symmetry of the system. Indeed, the complete master equation in Equation A.7 does connect different $m_j$ subspaces through the refilling term. Since we have shown this term to be negligible for the dynamics within a given subspace, we can exploit conservation of $m_j$ to analyze its subspaces separately, and have done so in the main text by fixing $m_j = 1/2$.

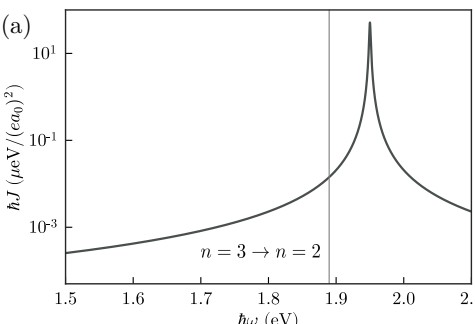
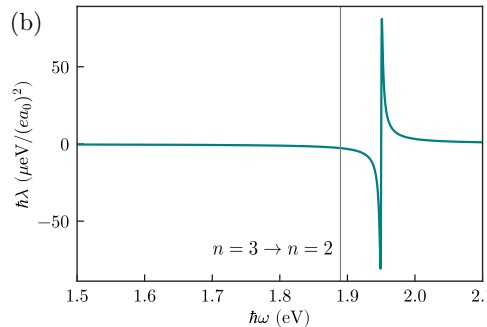

Figure 3: (a) Model spectral density, $J(\omega)$. (b) Integral of the spectral density that appears in the shift, $\lambda(\omega)$.

## C  Numerical check

In order to verify the validity of the derived Lindblad equation and effective Hamiltonian, we here apply it to a simplified system for which an exact solution is possible. To do so, we study the populations of the states with $n = 3$ in the hydrogen atom coupled to an electromagnetic bath whose spectral density is a Lorentzian. The density and the corresponding energy shift integral are shown in Figure 3a and Figure 3b, and are given by

$$J(\omega) = \frac{g^2}{\pi} \frac{\kappa/2}{(\omega - \omega_M)^2 + (\kappa/2)^2}, \tag{C.1}$$

$$\lambda(\omega) = \int_{-\infty}^{\infty} d\omega' \frac{J(\omega')}{\omega - \omega'} = g^2 \frac{\omega - \omega_M}{(\omega - \omega_M)^2 + (\kappa/2)^2}, \tag{C.2}$$

with parameter values $\hbar g = (9/\sqrt{5}) \cdot 10^{-4}$ eV/$(ea_0)$, $\hbar\kappa = 2 \cdot 10^{-3}$ eV and $\hbar\omega_M = 1.95$ eV. It is well-known that a Lorentzian spectral density is completely equivalent to a single mode coupled to a completely flat, i.e. Markovian, bath [43,44], with dynamics described exactly by a Lindblad equation [45],

$$\dot{\rho} = -\frac{i}{\hbar} \left[ H_{at} + \hbar\omega_M a^\dagger a, \rho \right] - \frac{\kappa}{2} \{a^\dagger a, \rho\} + \kappa a \rho a^\dagger, \tag{C.3}$$

where $a$ is the bosonic annihilation operator of the bath mode. Hence, we can compare the approximate solutions obtained with our approaches, Equation A.7 and Equation B.4, to the exact dynamics given by Equation C.3. We take $|\psi(0)\rangle = |n = 3, l = 0, j = 1/2, m_j = 1/2\rangle |n_{ph} = 0\rangle$ as the initial state and propagate it in time.

In the exact calculations, we include the first 4 Bohr levels of the hydrogen atom with their complete fine structure (60 states), which gives converged results. The population of the $|3, l, j, 1/2\rangle$ states are plotted in Figure 4. The three largest populations are perfectly well described by both our Lindblad equation and the effective, non-Hermitian Hamiltonian. It should be noted that the dynamics calculated using the BR approach are essentially the same as those obtained from the Lindblad equation and therefore not shown separately. There are two additional lines that are present only in the exact dynamics and the Lindblad equation, with populations of the order of $10^{-3}$. These are states that become populated through the refilling terms within the $n = 3$ subspace discussed above. These are unrealistically large here because the spectral density chosen here to enable comparison with an exact result does not obey the physical constraint $J(\omega) = 0$ for $\omega \leq 0$. In contrast, the spectral density used in the main text obeys these physical constraints and the refilling term can indeed be discarded with much less impact.

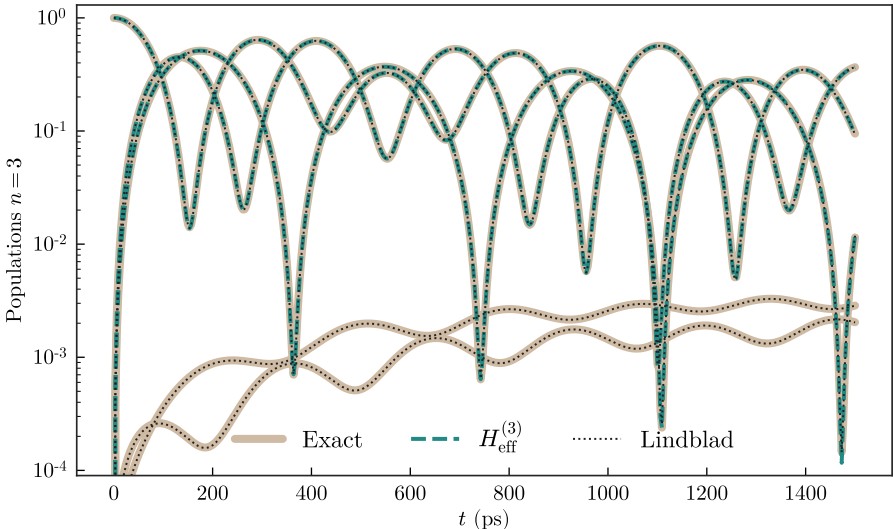

Figure 4: Time evolution of the atomic populations. Thick brown lines: numerical solution to the exact dynamics (Equation C.3). Green dashed lines: effective Hamiltonian (Equation B.4). Black dotted lines: Lindblad master equation (Equation A.7).

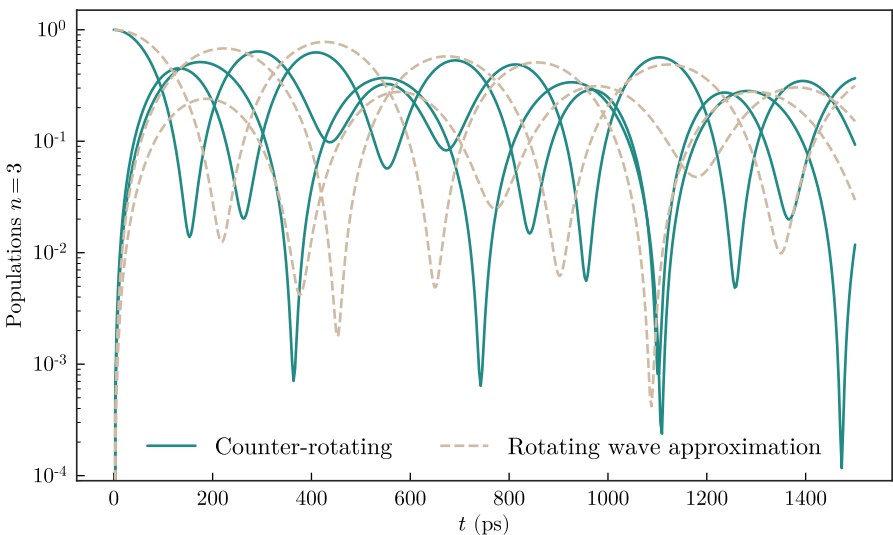

Figure 5: Dynamics calculated with the effective Hamiltonian. Green lines: including the contribution of the CR terms. Brown dashed lines: the rotating wave approximation has been performed on the light-matter Hamiltonian.

Having checked the validity of Equation B.2, we may use it to reinforce the claim that the CR terms are important in our work. In Figure 5, we compare the dynamics when the effect of the CR terms is included and when it is not due to the rotating wave approximation. Clearly, the marked differences in the oscillations indicates that the CR terms significantly contribute to the CP shift and thus to the dynamics.

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
