# Peer review of "Vacuum-field-induced state mixing"

_SciPost Physics, doi:SciPost Phys. 15, 252 (2023)_

## Round 1 · Referee Report · Anonymous (Referee 1) · 2023-7-18

Report

The authors describe a set-up in which the Casimir-Polder interaction is capable to induce mixing between otherwise (near-)degenerate quantum states. The example chosen is the fine structure splitting of the hydrogen atom for which the expected Casimir-Polder shift is of the same order of magnitude as the level splitting itself. This effect is somewhat reminiscent of the van der Waals interaction between highly excited (Rydberg) atoms where the interaction can no longer be treated as a perturbation, and an exact diagonalization of the full Hamiltonian has to be performed [e.g. T.G.Walker and M.Saffman, PRA 77, 032723 (2008)]. I would like the authors to comment on the possible relation of their work to Rydberg physics. In a similar context, Casimir-Polder-induced state mixing has already been considered [J.Block and S.Scheel, PRA 100, 062508 (2019)] in the context of surface-induced macrodimers between Rydberg atoms. It has even been shown that Casimir-Polder interaction can induce Rabi oscillations between degenerate states [M.Donaire, M.-P.Gorza, A.Maury, R.Guerout, and A.Lambrecht, EPL 109, 24003 (2015)]. It is therefore incorrect to claim that '...this interaction remains unexplored'.

As for some detailed questions and comments: 1) lines 44/45: There have been a number of non-perturbative approaches to dispersion interactions, notably [S.Y.Buhmann, L.Knoell, D.-G.Welsch, and H.T.Dung, PRA 70, 052117 (2004); S.Y.Buhmann and D.-G.Welsch, PRA 77, 012110 (2008)]. 2) line 125: The reference to Appendix A is somewhat inconsistent, as Appendix A itself refers to Eqs.(6a) and (6b) on the following page. 3) Eq.(5): In what way is the appearance and structure of H_CP similar or different to the Lamb shift renormalization in Ref.[11], Eq.(3.140)? 4) line 167: I believe that such avoided crossings have been discussed in the aforementioned Rydberg macrodimer work. 5) Figure 2: It is perhaps helpful to the (intended?) atomic physics community if the energy units where given in MHz rather than eV. 6) line 320: What is the rationale behind and the effect of introducing the geometric mean?

In conclusion, I believe the manuscript provides an interesting example of mixing of (near-)degenerate states by body-mediated vacuum fluctuations. It is generally well written, and deserves publication in SciPost, after suitable revision.

  • validity: -
  • significance: -
  • originality: -
  • clarity: -
  • formatting: -
  • grammar: -

Author:  Diego Fernández de la Pradilla  on 2023-10-09  [id 4031]

(in reply to Report 1 on 2023-07-18)

We thank the referee for the detailed report, the suggestions for improvement, and for bringing various pertinent articles to our attention. We have addressed all the points raised by the referee, and we believe that the paper will be significantly improved as a result. In the following, we first address the general comments and then the numbered questions and comments.

I would like the authors to comment on the possible relation of their work to Rydberg physics.

Although the system explored in our paper involves a hydrogen atom, Rydberg atoms are indeed a very natural extension of the work. The physical consequences of Rydberg-Rydberg interactions make them a platform of great interest that would provide additional phenomenology to investigate. In our work, we focus on the off-diagonal Casimir-Polder couplings within a single atom to isolate them and highlight their relevance. However, we fully agree that the combination of the effect described in our paper with Rydberg interactions is an interesting avenue to explore. In the first paper brought up by the referee, [Walker and Saffman, PRA 77, 032723 (2008)], an effective second-order Hamiltonian is derived with non-diagonal terms that cannot be neglected, like in our work. The work bears some resemblance to our paper (atomic systems, electromagnetic interactions, state mixing), but with a number of important differences: (i) our physical system is very different, being comprised of a dielectric nanoparticle and an atom, rather than two atoms interacting with each other. While a second atom could be seen in some sense as the ultimate limit of a ''nanoparticle'', there are important conceptual differences. For example, the nanoparticle supports a continuum of modes, instead of a few discrete ones as the atom, such that not only energy shifts, but also incoherent processes (losses) become relevant, necessitating an open-quantum-systems treatment. (ii) In our work, we derive a master equation that conveniently contains the relevant information including both energy shifts and couplings as well as losses induced by the environment and is applicable to many systems. (iii) The focus and thus also the conclusions of our work are very different from that of Walker and Saffman, as they are mainly concerned with blockade physics. We have referenced the article in the resubmitted version since it provides relevant context for our research.

Regarding the claim "... this interaction remains unexplored"

The second reference provided by the referee [Block and Scheel PRA 100, 062508 (2019)], does include Casimir-Polder shifts and describes state mixing. However, there are again a number of important differences to our work. (i) The system is quite different. Like Walker and Saffman, they study mixings in atomic pairs, adding a perfectly conducting half-space that is responsible for affecting the mixing in the 2-atom system. In our case, a realistic nanoparticle is responsible for the mixing of 1-atom states individually, and without breaking rotational symmetry. (ii) Block and Scheel do not derive a master equation and do not consider decay, while it is essential in our description. (iii) The physical consequences of the anticrossing features are also different: they obtain a potential well giving rise to vibrational states of the 2-atom system. Hence, although Block and Scheel study a different system with different techniques and explore different physics, we believe the article adds important context to our research, and we now cite it in the resubmitted version. As for the third paper mentioned by the referee, [Donaire, Gorza, Maury, Guerout and Lambrecht, EPL 109, 24003 (2015)], it is much closer conceptually to our work than the previous two. Nevertheless, there are some important methodological differences: (i) we develop a master equation that accounts for the decay dynamics, while they use a unitary evolution approach to obtain the effective dynamical parameters. (ii) Their study is limited to a 2-level subspace where the coupling depends on an average frequency, while ours provides a general approach that can deal with arbitrarily many levels. (iii) The inclusion of many levels also means that "mixing" can take many forms. We explore and explicitly quantify this through the participation ratio. We have cited the article in the resubmitted version. We thank the referee for bringing these references to our attention. Although we still believe that our work provides a new approach and gives access to physics that have not been studied in detail, we have rephrased our previous claim that "... this interaction remains unexplored" to more precisely represent the current status of the literature regarding this physical effect.

As for the numbered questions and comments:

  1. We thank the referee for making us aware of these references. They are indeed relevant and included in the resubmitted version.
  2. As suggested by the referees, we have rewritten the Methods section to clarify and improve its readability. To achieve this, we have restructured the materials presented in the Methods and the Appendix sections.
  3. Although the $H_{CP}$ appears in a similar way as the Lamb shift in the book by Breuer and Petruccione, there is a key difference in how it is obtained. Specifically, Eq. (3.140) in the book is obtained through the secular approximation and consequently, lacks the off-diagonal couplings between non-degenerate states. One of the main merits of our paper is the inclusion of precisely those terms in a way that also a Lindblad equation is obtained, which thus allows us to go beyond the limitations of the secular approximation.
  4. We agree that there are conceptual similarities, although also some clear differences: the crossings investigated in the reference given by the referee are 2-atom energies, conceptually similar to molecular potentials that become modified because the electromagnetic interactions between the atoms are affected by the presence of a nearby macroscopic body. In contrast, we study interactions between the states within a single atom which are modified by the presence of the nanoparticle. We have modified the statement in the resubmission to provide this context.
  5. We have included a MHz scale in the resubmitted version for improved clarity.
  6. Once the BR equation is derived, we look for a procedure that faithfully approximates the relevant off-diagonal terms and still allows for a refactoring such that a Lindblad master equation emerges. We have found, confirming the results by [McCauley, Cruikshank, Bondar and Jacobs, npjqi 6, 74 (2020)], that the geometric mean is a great candidate. As an example, if one were to use an arithmetic mean instead, the decay part would not satisfy the requirements of a Lindblad equation because it would lead to negative decay rates, and thus a dynamics that is not completely positive. We have clarified this in the resubmission.

---

## Round 1 · Referee Report · Anonymous (Referee 2) · 2023-9-1

Report

The paper by Fernandez de la Pradilla et al discusses vacuum-induced effects, where eigenstates of an atomic Hamiltonian can become hybridized due to the interaction with the electromagnetic field. The authors propose a setup where this mixing effect can be observed, leading for instance ot a substantial modification of the decay rate of atomic states. The results are interesting: the authors identify a setup that could allow to measure this effect, which has been extensively discussed in the literature. The presentation is generally clear. The paper deserves to be published after the authors have carefully considered the points reported below.

1) Missing reference to relevant literature. The phenomenon that the authors denote by vacuum-induced state mixing has been extensively discussed in the literature for atoms and molecules in free space, see for instance “Steady-state quantum interference in resonance fluorescence” by D A Cardimona, et al, J Phys B 15, 55 (1982) and “Spontaneous radiative coupling of atomic energy levels” D. A. Cardimona et al, Phys. Rev. A 27, 2456 (1983) and the book Z. Ficek et al “Quantum interference and coherence” (Springer ed). The setup the authors propose (Hydrogen atom close to an aluminiun nitride nanoparticle) has -to my knowledge- not been discussed before. Nevertheless, the claim that “the interaction remains unexplored” (see abstract and main text) is incorrect and shall be accordingly revised.

2) Free-space and scattering contributions of the Green tensor. In the derivation of the master equation the authors neglect the off-diagonal contribution of the free-space Green tensor. However, the free-space contribution gives also rise to state-mixing (see literature mentioned above). Moreover, in the presence of the scattering contribution, in the master equation there should be also a term corresponding to an interference between free-space and scattering term of the Green tensor that shall modify both the imaginary part of the non-Hermitian Hamiltonian as well as the jump term. The authors shall discuss what is its order of magnitude and when can it be neglected.

3) Model. In general, Section 2 is difficult to follow without reading the appendix (for instance, the discussion after Eq. (6b) remains vague without specifying what shall be the raising and lowering operators there mentioned). The clarity of the presentation could improve if the Appendix would be included in the “Methods” section.

4) Master Equation (5). This master equation is obtained from the Bloch-Redfield equation in the appendix, which the authors claim to take from Ref. [11] and from the book C. Cohen-Tannoudji et al “Atom-Photon interactions”. Equation (5) is then derived from the Bloch—Redfield after applying the Markov-approximation and after symmetrizing by hand the terms which lead to state-mixing. As far as it concerns the Markov approximation: The authors claim that Eq. (5) is valid at zero temperature. This cannot be correct, since at zero temperature the Markov approximation becomes invalid. As far as it concerns the symmetrization: The authors do so by applying a kind of geometric mean. Is this procedure the same as the procedure described in the book of Z. Ficek et al Chapter 2 “Master equation of a multi-dipole system”? If not, how does it differ from it?

5) Partial secularization. In general, the criterion applied by the authors -when performing what they denote by “partial secularization”- is based on approximate considerations, that consist of comparing the energy difference of transitions which can be coupled with their linewidth. While this can be ok for the specific case they consider, where they can restrict their analysis to a well defined manifold, the procedure is then very difficult to generalize to other cases. Can the authors provide a more rigorous criterion?

6) Appendix, positive semidefiniteness of the Lindblad operator, discussion before Eq. (15). It is difficult to understand what the authors mean when they write “we are not aware of a general proof of positive semidefiniteness for arbitrary spectral densities”. The sentence seems contradictory: in fact, the Markov approximation can be applied provided the characteristic time scale of the reservoir’s autocorrelation function is sufficiently small, so that the corresponding spectrum can be assumed to be flat. In that case the specific form of the spectral density becomes irrelevant. Then, for positive semidefiniteness it should be sufficient to check that the master equation the authors derived fulfils the Lindbald theorem.

7) Non-Hermitian Hamiltonian. The sentence at the beginning of 2.2: about the Bloch-Redfield equation: “… but leads to non-trace-preserving dynamics in which the coherent evolution cannot be interpreted as an effective non-Hermitian Hamiltonian”. This needs some explanation. In fact, non-Hermitian Hamiltonians lead to non-trace-preserving dynamics.

8) Subradiant state. In the results section, the authors report a decrease of the decay rate due to state mixing and denote the corresponding state by “subradiant state”. A different wording (maybe “metastable state”?) would avoid confusion with the collective phenomenon of subradiance.

  • validity: good
  • significance: good
  • originality: good
  • clarity: good
  • formatting: reasonable
  • grammar: excellent

Author:  Diego Fernández de la Pradilla  on 2023-10-09  [id 4032]

(in reply to Report 2 on 2023-09-01)

We thank the referee for the careful report, the suggestions for improvement, and for bringing various pertinent articles to our attention. We have addressed all the points raised by the referee, and we believe that the paper will be significantly improved as a result. In the following, we address referee’s comments.

  1. We thank the referee for pointing out these references. Indeed, they provide an important and solid conceptual basis for our study, and we were not aware of them. We have cited them in the resubmitted version, and have modified the statement “the interaction remains unexplored” to make it more accurate.
  2. Such free-space effects are negligible within the subset of states considered in our work, where the fine structure determines the energetic scale. We first note that in the initial Bloch-Redfield equation, there are no interference terms and the effect of the free space and scattered contributions are additive. The geometric mean used in our procedure to obtain a Lindblad equation does then introduce something akin to an interference term, but closer inspection reveals that it does not follow the usual intuition of interference enhancing the effect of small contributions. In particular, using that $\lambda_i=\lambda_{i,0}+\lambda_{i,s}$ is the sum of free-space and scattered contributions, the geometric mean can be expanded to first order in $\lambda_{i,0}$ as $\sqrt{\lambda_1\lambda_2}\simeq \bar{\lambda}_s + \bar{\lambda}_{s}\Big(\lambda_{1,0}/\lambda{1,s}+\lambda_{2,0}/\lambda_{2,s}\Big)/2$, where $\bar{\lambda}_s = \sqrt{\lambda_{1,s}\lambda_{2,s}}$. As discussed in the article, the terms that are most relevant in the dynamics fulfill $\lambda_{1,s}\simeq\lambda_{2,s}\simeq\bar{\lambda}_s$, such that the correction due to free-space terms is still an additive correction corresponding to the (arithmetic) mean of the free-space contributions with prefactors that are close to one. Closer inspection reveals that these can also be neglected. The relevant orders of magnitude are the following: The fine structure is$ O(10^{−6})$ eV, the nanoparticle-induced energy shift reaches and surpasses the same order of magnitude, and the free-space Lamb shift for the 7s state is $O(10^{−7})$ eV, while for states with higher angular momentum it is at most $O(10^{−9})$ eV (as discussed in, e.g., Bethe and Salpeter, “Quantum Mechanics of One- and Two-electron atoms”). This approximation might induce a slight inaccuracy for s states (due to them possessing the largest Lamb shifts), but for states with $l > 0$ it is quite precise.
  3. As suggested by the referees, we have rewritten the Methods section to clarify and improve its readability. To achieve this, we have restructured the materials presented in the Methods and the Appendix sections.
  4. We thank the referee for raising this point. We have replaced the “zero temperature” comment with a clarifying statement regarding temperature in the paper. It is more accurate to say that we assume the thermal energy $k_BT$ to be sufficiently low compared to the transition frequencies that we can neglect terms proportional to $n_T (\omega_i)$, the average number of photons of frequency ωi in a thermal bath of temperature $T$. For the relevant transitions in this work, with energies on the order of $0.1$ eV, this is well-justified even at room temperature (where $k_BT \simeq 26$ meV). For that reason, we can neglect the thermal contributions proportional to $n_T (\omega_i)$. In the book by Ficek and Swain, a master equation is derived for multilevel atoms that includes off-diagonal couplings, limited to free-space or Fabry-Perot cavities. In the derivation, it is implicitly assumed that the relevant frequencies are close enough that one can replace $\gamma_i$ and $\gamma_j$ by $\sqrt{\gamma_i\gamma_j}$. This point is also addressed in [Buchheit and Morigi, PRA 94, 042111 (2016)], at the end of section II B. Indeed, for the decay, this approximation is the same as ours. However, the final master equation, eq. (2.49) of the book by Ficek and Swain differs from ours in its treatment of the Lamb shift, where we use essentially the same approximation again to recover the Lindblad form, while they do not. In fact, their expression of the Lamb shift operator given by the last two lines is not Hermitian, because the off-diagonal terms are not complex conjugates of each other unless $\omega_i=\omega_j$. This is due to the integrand of $\delta^{(±)}_{ij}$ depending on $\omega_j$ . Nevertheless, it is true that at least part of the spirit behind the usage of the geometric mean is present in the book, and we have cited it accordingly.
  5. We argue that the criterion is indeed rigorous. The criterion is that the size of the environment-induced coupling must be sufficiently small relative to the frequency difference of the transitions (where “sufficiently” is determined by the desired level of accuracy), as discussed in the manuscript. Of course, the details of this comparison are highly system-dependent, and only in cases (like the present one) where the energy levels are organized in well-separated groups will this allow removal of a whole class of terms, but the relevant quantities to compare to assess the validity of the secular approximation for each term are well-defined. We furthermore mention that the partial secularization is not required to reach any of the conclusions presented in our work, but it simplifies the equations significantly.
  6. We thank the referee for making us notice that this point needed further clarification. We have improved the phrasing according to the following. A master equation is of Lindblad type if it has a certain mathematical structure and the associated decay matrix (also known as Kossakowski matrix) is positive semi-definite. Our equation does indeed have the right structure, but it is not clear a priori whether the corresponding Kossakowski matrix, $\tilde{\Gamma}$, is positive semi-definite or not. This Kossakowski matrix contains information about the spectral density of the bath and the emitter’s level structure in an intricate way, as a result of the manipulations described in the appendix of the initial submission. For spectral densities describing bath operators with vanishing cross-correlations, we do know that it will be positive semi-definite, as the environment can then be interpreted as a collection of independent baths that can be treated separately (and in this case, positive-semidefiniteness can be shown explicitly). In the system we treat in the current manuscript, the cross-correlations are zero because the Green tensor is diagonal, and our master equation is thus of Lindblad type. Whenever cross-correlations appear, however, it is not guaranteed that the matrix will be positive semi-definite, and we have indeed found cases where it is not. Regarding the Markov approximation and the flatness of the spectrum, a more precise statement is that each transition’s ($i$) associated decay rate is given by $J(\omega_i)$, as though the spectrum where flat and equal to its value at $\omega_i$. However, since there are many transitions with very different frequencies, it is not really true that the specific form of the spectral density is irrelevant. For the particular sentence in our paper addressed by the referee, it is accurate to say that the possibility of non-positivity of the Kossakowski matrix comes from the emergence of bath cross-correlations. These cross-correlations are measured by the off-diagonal terms of the spectral density tensor, and that is what we meant by “arbitrary spectral densities”.
  7. We have clarified this statement in the resubmitted version. We first note that the Bloch-Redfield equation does indeed preserve the trace. What we meant by “the coherent evolution cannot be interpreted as an effective non-Hermitian Hamiltonian” is based on the trajectory/quantum jump interpretation of Lindblad master equations. In this interpretation (and its numerical implementation through quantum Monte Carlo simulations), the evolution of a density matrix under a Lindblad equation can be understood as an average over many quantum trajectories, which correspond to individual realizations of the “experiment”. In such a picture, the system evolves coherently under a non-Hermitian Hamiltonian for a certain time interval, until it randomly “jumps” according to the jump term of the master equation. The probability for such jumps is proportional to the prefactor accompanying the jump term, that is, the decay rates. However, when the dynamics is not completely positive (as in the standard Bloch-Redfield equation), some of those rates are negative, which then does not allow for the same interpretation of coherent non-Hermitian dynamics with quantum jumps as obtained with a Lindblad equation.
  8. We believe that there is a close connection between the physics behind “collective subradiance” and our effect. In the first, interference effects between dipoles at different positions yields the phenomenon, and in ours it is the interference between several dipolar transitions within the same atom. The fascinating aspect is that the same vacuum-field-induced interactions that generate the decay in our case also lead to the state mixing that allows for the formation of the subradiant state. We clarify this and include the “metastable” label as well to more accurate convey the point.

---

## Round 2 · Referee Report · Anonymous (Referee 2) · 2023-11-18

Report

The authors have convincingly replied to the points raised in the report.

Requested changes

Two points, extensively discussed in the reply letter by the authors, are missing in the manuscript : i) the comment on the fact that vacuum-induced interference in free space is negligible in the setting the authors consider ii) the reference to Buchheit et al, PRA (2016), to which the authors referenced in their reply letter when justifying the geometric mean.

---

## Round 2 · Author Response

We thank the referees for their comments and suggestions. We have addressed their comments and improved the manuscript accordingly. We trust that our work is now ready for publication in SciPost Physics.

Please see below for a list of changes. The detailed replies to the referees have been posted in the first submission.

---

## Round 2 · List of Changes

• Improvement of explanations and inclusion of references suggested by the referees (more details in the replies).
  • The Methods section has been restructured to include part of what was previously in the Appendix, to improve the clarity and exposition.

---

## Editorial Decision

published